# Genome-Wide Analysis of Long Non-Coding RNAs Related to UV-B Radiation in the Antarctic Moss *Pohlia nutans*

**DOI:** 10.3390/ijms24065757

**Published:** 2023-03-17

**Authors:** Shuo Fang, Bailin Cong, Linlin Zhao, Chenlin Liu, Zhaohui Zhang, Shenghao Liu

**Affiliations:** 1Key Laboratory of Marine Eco-Environmental Science and Technology, First Institute of Oceanography, Ministry of Natural Resources, Qingdao 266061, China; 2Laboratory for Marine Ecology and Environmental Science, Pilot National Laboratory for Marine Science and Technology (Qingdao), Qingdao 266061, China

**Keywords:** lncRNA, bryophytes, flavonoids, metabolome, transcriptome, UV-B

## Abstract

Antarctic organisms are consistently suffering from multiple environmental pressures, especially the strong UV radiation caused by the loss of the ozone layer. The mosses and lichens dominate the vegetation of the Antarctic continent, which grow and propagate in these harsh environments. However, the molecular mechanisms and related regulatory networks of these Antarctic plants against UV-B radiation are largely unknown. Here, we used an integrated multi-omics approach to study the regulatory mechanism of long non-coding RNAs (lncRNAs) of an Antarctic moss (*Pohlia nutans*) in response to UV-B radiation. We identified a total of 5729 lncRNA sequences by transcriptome sequencing, including 1459 differentially expressed lncRNAs (DELs). Through functional annotation, we found that the target genes of DELs were significantly enriched in plant-pathogen interaction and the flavonoid synthesis pathway. In addition, a total of 451 metabolites were detected by metabonomic analysis, and 97 differentially change metabolites (DCMs) were found. Flavonoids account for 20% of the total significantly up-regulated metabolites. In addition, the comprehensive transcriptome and metabolome analyses revealed the co-expression pattern of DELs and DCMs of flavonoids. Our results provide insights into the regulatory network of lncRNA under UV-B radiation and the adaptation of Antarctic moss to the polar environments.

## 1. Introduction

UV light is classified into UV-C (100–280 nm), UV-B (280–315 nm) and UV-A (315–400 nm), according to the different wavelengths [1]. UV-B is the most dangerous part of UV radiation, which can produce various harmful effects on organisms [2]. Plants perceive UV-B as an environmental signal and a potential abiotic stress factor that affects development and acclimation [3]. Since the 1980s and 1990s, the reduction of the stratospheric ozone layer has resulted in an increase in UV-B light on Earth’s surface [4]. The Montreal Treaty is committed to reducing polar ozone depletion. The ambient level of UV-B radiation is affected by latitude, season, biomass burning aerosol and the angle of the sun. UV-B levels also depend on elevation, cloud cover, surface reflectance, and vegetation canopy thickness [5]. At high latitudes, the intensity of UV-B light is considered to be more affected by stratospheric ozone recovery, cloud and surface reflectance [6]. Particularly, the biomass-burning aerosol produced by the Australian “Black Summer” wildfires in 2019–2020 December–January–February (DJF) will speed up and then postpone the collapse of the polar vortex, which, in turn, will prolong the ozone hole that was detected in 2020 [7]. In 2020, the Antarctic ozone hole still led to record-breaking increases and had extended to most areas of the Antarctic continent according to a report by the World Meteorological Organization [8].

Long non-coding RNAs (lncRNAs) are transcripts with a length of more than 200 nucleotides and no or limited protein-coding ability [9,10]. They are divided into long intergenic ncRNAs (lincRNAs), intron lncRNAs (incRNAs), antisense RNAs and natural antisense transcripts (NATs). LncRNA forms the molecular framework of macromolecular complexes to expression, by complementing with RNA sequences or homologously regulating genes [11]. In this process, target genes can be regulated by lncRNA in both cis- and trans-acting ways [12]. Cis-acting elements are involved in the expression regulation or transcriptional activation of mRNA close to lncRNA, while trans-acting elements can modify genes at several distant places by attaching to transcriptional enhancers or promoters. In mammals, lncRNA research has made several great progresses. Recently, the studies of plant lncRNAs have also gradually caught up to that of mammals [13], and large amounts of lncRNAs have been found in the genomes of different plant species [14,15,16]. In plants, lncRNAs play a role in gene silencing, root organogenesis, seedling photomorphogenesis and reproduction [17,18,19], while they are also involved in regulating salt stress, drought stress and cold stress [20,21,22]. However, there are few studies on the roles of lncRNA involved in regulating plant responses and signal pathways to UV radiation.

In the Antarctic terrestrial ecosystems, plants are experiencing high UV-B radiation of 3.4–6.2 mW/cm^2^ [23]. On the land of Antarctica, lichens and bryophytes are the main vegetation. Under the natural UV radiation, the moss (*Bryum argentium*) and lichen (*Umbiliaria apina*) contain UV-B absorbing compounds, and the contents of phenols and total carotenoids have significant changes [24,25,26]. In addition, the Antarctic moss, *Leptobryon pyriforme*, can resist continuous UV-B radiation through the flavonoid biosynthesis pathway, jasmonate signaling, UVR8-mediated signaling and DNA repair system [27]. Meanwhile, the distribution area of *Colobanthus quitensis* and *Deschampsia antarctica*, the only two vascular plant species native to Antarctica, are increasing due to climate change in Antarctica. They are restricted to seasonally or permanently snow- and ice-free areas accounting for ~3.4% of the Antarctic continent, where sufficient summer snowmelt occurs [28]. The chlorophyll content of *C. quitensis* and *D. antarctica* decreased under UV-B radiation, leading to the reduction of their vegetative growth, whereas the content of UV-B-absorbing compounds was significantly increased [29]. It is suggested that these Antarctic terrestrial plants can resist UV-B radiation by synthesizing antioxidants (including UV-B absorption pigments, flavonoids and anthocyanins) that function as an effective damage repair system [24,30]. However, transcriptional regulation and metabolic features, as well as adaptation processes to UV-B radiation remain poorly understood in these Antarctic terrestrial plants.

In this study, the transcriptome analysis in *Pohlia nutans* under UV-B radiation was performed by high-throughput sequencing technology. We predicted the lncRNAs from transcripts based on the transcript length screening and coding potential prediction. A total of 1459 differentially expressed lncRNAs (DELs) were obtained and their target genes were predicted by cis- and trans-regulated modes. Moreover, the target genes of DELs were summarized into different categories, including plant-pathogen interaction, phenylpropanoid biosynthesis, arachidonic acid metabolism, flavonoid biosynthesis, and fatty acid elongation. In addition, we identified 97 differentially changed metabolites (DCMs) using the widely targeted metabolomics technique, which were mainly distributed in the flavonoid biosynthesis pathway. Integrated multi-omics analysis highlights that the flavonoid pathway and plant-pathogen interaction regulated by DELs might play a major role in the process of Antarctic moss, *P. nutans*, resistance to UV-B radiation.

## 2. Results

### 2.1. Identification and Characterization of lncRNAs in Pohlia nutans

We constructed six cDNA libraries from the control group (CK1, CK2, and CK3) and UV-B treatment group (UV-B1, UV-B2, and UV-B3), respectively. Transcriptome sequencing generated 537,430,626 raw reads in total, with more than 70,000,000 raw reads per sample. The number of clean reads from the CK1, CK2, CK3, UV-B1, UV-B2, and UV-B3 samples, after filtering the raw reads, was 79,469,444, 98,500,292, 88,589,850, 86,334,606, and 95,201,204, respectively. The Q_20_ values of clean reads were greater than 96%. Q_30_ scores were greater than 89.9% in all samples. The contents of GC ranged from 43.84 to 45.38% (Table 1). We have submitted the RNA sequencing data generated in our study to the National Genomics Data Center (NGDC, https://ngdc.cncb.ac.cn, accessed on 12 July 2022) under the accession number of CRA006053. Then, the clean reads were aligned to the *P. nutans* reference genome, by using HISAT2 [31], and approximately 49.47–63.75% of the clean reads were mapped to the reference genome (Appendix A). These values indicated a high quality of clean reads.

We analyzed the correlation of the parallel samples to test the reliability of the experiment. The similarity of expression patterns among samples increased as the Pearson’s correlation coefficient approached 1.0 (Appendix A). Through principal component analysis (PCA), we analyzed the components of score plots (PC1 and PC2), calculating 64.44%, and 12.35%, respectively. PCA results showed that there were significant differences between CK and UV-B, and that they clustered well within the group (Appendix A). LncRNAs were predicted from the transcripts based on the transcript length screening and coding potential prediction, obtaining mRNA and lncRNA information at the same time. We compared the mRNA and lncRNA sequence features that were generated through transcriptome sequencing analysis. The sequencing characteristics of the identified lncRNA and mRNA were compared. The distribution curves of exon number and transcript length were plotted including lncRNAs and protein-coding transcripts, respectively. All lncRNAs contained less than eight exons. In general, 91.16% of mRNA had more than two exons, while 63.12% of lncRNA had one or two exons (Appendix A). LncRNAs with lengths of 300–700 bp accounted for 44.63% of the identified lncRNAs, while mRNAs larger than 2000 bp account for 78.71% of total mRNAs (Appendix A). The mean length of mRNA sequence was extremely larger than that of lncRNA sequence. The number of predicated lncRNAs and mRNAs varies among different chromosomes (chr1–chr22, Figure 1A,B), while they have similar rise or decline trends about the number on the chromosomes. After removing the sequence of scaffold, lncRNA and mRNA were most abundant on chromosome 2 (chr2).

### 2.2. Prediction, Classification and Expression Profiles of lncRNAs in Pohlia nutans

The softwares CPC2, CNCI and PLEK were used to predict the non-coding transcripts, respectively. A total of 5729 transcripts were identified as lncRNAs through cross validation of these three methods (Appendix A and Appendix A). Of them, 2131 lncRNAs (37.2% of the total) were located in intergenic regions and were classified as lincRNAs; 2538 lncRNAs (44.3% of the total) that were located in the intronic regions of coding genes were classified as intronic lncRNAs. The remaining 1060 transcripts were antisense-lncRNAs, accounting for 18.5% of the total lncRNAs (Figure 1C). In addition, a total of 1459 differentially expressed lncRNAs (DELs) were determined by comparing the *p*-values and fold-change. The hierarchical clustering of a heatmap using the FPKM value showed that the DELs of samples from the CK or the treatment group were clustered together, respectively. These two groups were clearly separated (Appendix A). Volcano plots were obtained by analyzing DELs between the two groups. When compared with the CK, we found that 954 DELs were significantly up-regulated and 505 DELs were significantly down-regulated (Figure 1D and Appendix A). According to the results of transcriptome analysis, we obtained 19,045 differentially expressed genes (DEGs), of which 9518 were up-regulated and 9527 were down-regulated (Appendix A). Therefore, our results showed that a large number of DELs and DEGs were up-regulated in response to UV-B radiation of *P. nutans*.

### 2.3. Functional Annotation of Cis- and Trans-Target Genes Involved in Flavonoid Biosynthesis and Plant-Pathogen Interaction Pathways

After screening DELs according to the experimental purpose, we first screen the target genes, and then conduct enrichment analysis. Enrichment analysis was conducted to study the distribution of DEL target genes in GO (Gene Ontology) and KEGG (Kyoto Encyclopedia of Genes and Genomes), in order to clarify the function of DELs in the experiment. LncRNA can act on additional genes through base complementary pairing, and thus control the expression of genes that are nearby. We predicted the biological function of lncRNA by its location relationship (cis) with protein-coding genes. First, we performed GO functional annotation of DEL cis-regulated target genes. In biological processes category, the largest number of genes are enriched in inorganic ion homeostasis (GO:0098771); in cellular components category, genes were mainly enriched in membrane-associated annotations, which was similar to the enrichment result of mRNA; in the molecular function category, channel activity (GO:0015267), passive transmembrane transporter activity (GO:0022803) and oxidoreductase activity (GO:0016667) were the main categories of DEL cis-regulated target genes (Figure 2A). For the KEGG pathway enrichment analysis, we found that the DEL cis-regulated target genes were mainly classified into five categories, including plant-pathogen interaction, phenylpropanoid biosynthesis, arachidonic acid metabolism, flavonoid biosynthesis and fatty acid elongation (Figure 2B).

Trans prediction could predict the possible relationships through sequence alignment. We predicted the trans-target genes of DELs through correlation analysis or co-expression analysis of the expression of lncRNA and mRNA between the samples. Then, the lncRNA trans mRNA was enriched and analyzed. According to GO functional enrichment, most of the trans-target genes of DELs in biological process were annotated in inorganic ion homeostasis (GO: 0098771) and chloroplast organization (GO: 0009658). In terms of the category of molecular functions, enzyme regulator activity (GO:0030234) and oxidoreductase activity (GO:0016616) had the largest numbers of the trans-target genes of DELs (Appendix A). The KEGG pathway analysis showed that the pathways annotated by the trans-target genes of DELs were assigned to three main categories: biosynthesis of cofactors, biosynthesis of amino acids and RNA transport (Appendix A). We found that a large number of lncRNA cis-target genes and trans-target genes annotated plant-pathogen interaction and flavonoid biosynthesis.

Subsequently, we conducted GO functional annotation and KEGG pathway enrichment analysis to explore the roles of differentially expressed genes (DEGs) under UV-B radiation. For GO functional annotation, DEGs were mainly annotated in cellular glucan metabolic (GO:0006073) and chloroplast organization (GO:0009658) in the biological processes. In cellular components, DEGs were significantly enriched in the microtubule (GO:0005874), an intrinsic component of the plasma membrane (GO:0031226), and plastoglobuli (GO:0010287). In the molecular function category, UDP-glycosyltransferase activity (GO:0008194), tubulin binding (GO:0015631) and microtubule binding (GO:0008017) had the largest numbers of DEGs (Appendix A). For KEGG pathway enrichment analysis, DEGs were largely distributed into five pathways, including plant-pathogen interaction, flavonoid biosynthesis, plant hormone signal transduction, amino acid biosynthesis, and metabolism of alpha-linolenic acid (Appendix A). Taken together, the integrated analyses showed that plant-pathogen interaction and flavonoid biosynthesis were the common enrichment pathways of DEG and DEL target genes.

### 2.4. DELs in Plant-Pathogen Interaction

Through KEGG enrichment analysis, we had found that the main pathways were plant-pathogen interaction and flavonoid synthesis. We then screened the DELs involved in these pathways to construct expression networks. The high-tolerance phenotype of plants exposed to UV-B may increase their resistances to biological stresses such as pathogens and insects [32]. In our study, a total of 386 DEGs related to the plant-pathogen interaction were detected by transcriptome analysis, of which 143 were up-regulated (Appendix A). At the same time, we detected a large number of DELs involved in regulating plant-pathogen interaction and most of them were up-regulated in response to UV-B radiation of *P. nutans* (Figure 3A). The target genes of these DELs were nitric oxide synthase (*NOS*), WRKY transcription factor 1 (*WRKY1*), pathogenesis-related protein 1 (*PR1*), WRKY transcription factor 25 (*WRKY25*), enhanced disease susceptibility 1 protein (*EDS*), disease resistance protein RPM1 (*RPM1*), 3-ketoacyl-CoA synthase (*KCS*) and disease resistance protein RPS2 (*RPS2*) (Figure 3B and Appendix A). *PR1* had antifungal activity and was the main family of PR proteins induced by pathogens or salicylic acid [33]. In this study, all of DELs involved in the regulation of PR1 gene were significantly up-regulated. At the same time, most of DELs regulating *RPS2* and *RPM1*, which are plant disease resistance genes, were also significantly up-regulated (Figure 3B).

The accumulation of secondary metabolites in plants under UV-B radiation, such as polyphenols, alkaloids, and terpenoids, may be strongly associated to the enhancement of the plant-pathogen interaction pathway [34]. Polyphenols are a large group of phytochemicals. They can be classified as flavonoid and non-flavonoid molecules (phenolic acids, hydroxycinnamic acids, stilbenes, lignans, and tannins). Through widely targeted metabolomics analysis, we found that most of differentially changed metabolites (DCMs) were polyphenol compounds, including flavonoids (i.e., gallic acid, cyanidin 3-O-(6″-malonylglucoside), luteolin 7-O-glucoside, orobol, eriodictyol, isoscutellarein, and luteolin), phenolic acids (i.e., caffeoyl-p-coumaroyltartaric acid, feruloyl glucose, protocatechuic acid-4-glucoside, cimidahurinine 5-(2-hydroxyethyl)-2-O-glucosylohenol, and 2,5-dihydroxy benzoic acid O-hexside), and tannins (procyanidin B2, and procyanidin B3). The content of these compounds increased significantly under UV-B stress (Appendix A). Polyphenol compounds can significantly up-regulate the expression of rhpRS gene in *Pseudomonas syringae*, and significantly inhibit the transcription of T3SS regulatory gene, hrpL, and many T3SS effector protein genes [35]. Therefore, polyphenols are essential to help plants resist pathogen invasion.

In addition, we determined the expression profiles of several selected DELs under UV-B radiation using the qPCR technique. Results showed that the expression levels of these DELs were up-regulated under UV-B radiation. Their corresponding target genes were *NOS* (LNC_003241 and LNC_003575), *WRKY1* (LNC_003749), *WRKY25* (LNC_000165 and LNC_002877), *PR1* (LNC_000595, LNC_003258, LNC_001246 and LNC_002202), *EDS* (LNC_003677), *RPM1* (LNC_001572, LNC_002009, LNC_002147 and LNC_004214), *KCS* (LNC_000458 and LNC_001473), and *RPS2* (LNC_000344, LNC_00490, LNC_001287 and LNC_002323), which was consistent with the data in transcriptome sequencing (Figure 4).

### 2.5. Integrated Multi-Omics Analyses Highlight the Role of the Flavonoid Biosynthesis Pathway under UV-B Radiation

We conducted the widely targeted metabolomics analysis using the UPLC-MS/MS platform to identify the changes of metabolites in *P. nutans* after exposure to UV-B radiation. The DCMs were screened using the threshold of |log_2_ (foldchange)| ≥ 1 and VIP (variable importance in project) ≥ 1. We therefore found 97 DCMs in *P. nutans* under UV-B radiation, with 80 up-regulated and 17 down-regulated metabolites between the treatment and CK. Among the DCMs, amino acids, phenolic acids and flavonoids were the most abundant metabolites. In addition, flavonoids account for 20% of the total up-regulated metabolites (Figure 5A and Appendix A). The top 20 DCMs were uncovered in comparison to the UV-B treatment group with the CK, based on VIP scores (Figure 5B). Particularly, gallic acid (mws0024) that belongs to flavonols was the most significant metabolite, with a VIP score greater than 1.495.

In order to uncover the relevance between DELs and DCMs in *P. nutans* under UV-B radiation, we carried out the integrated analyses between transcriptome and metabolome. Firstly, the target genes of DELs were predicted by co-expression analyses between lncRNA and mRNA. A large number of DELs involved in regulating flavonoid synthesis were detected by transcriptome sequencing analysis in *P. nutans* under UV-B radiation. The target genes of these DELs were chalcone synthase (*CHS*), flavonoid 3′-hydroxylase (*F3′H*), flavonoid 3′, 5′-hydroxylase (*F3′5′H*), flavones synthase (*FNS*), flavonol synthase (*FLS*), caffeoyl-CoA O-methyltransferase (*CCoAOMT*), flavanone 4-reductase (*DRF*), chalcone synthase (*CHS*), isoflavone synthase (*IFS*), shikimate O-hydroxycinnamoyltransferase (*HCT*), 5-O-D-quinate 3′-monooxygenase (*C3′H*), cinnamate 4-monooxygenase (*C4H*) and chalcone isomerase (*CHI*) (Figure 5C and Appendix A). In addition, 117 DEGs related to flavonoid synthesis, and 83 DEGs, were up-regulated (Appendix A). Consequently, several flavonoid compounds belonging to flavone, flavanone, flavonol, dihydroflavonol and isoflavones, were significantly accumulated in response to UV-B radiation (Figure 5D). In particular, qPCR analysis confirmed that under UV-B radiation, these DELs involved in regulating flavonoid biosynthesis were significantly up-regulated. Their corresponding target genes were *C4H* (LNC_000398, LNC_000980, LNC_003677 and LNC_004098), *CCoAOMT* (LNC_000418), *CHI* (LNC_002393 and LNC_002724), *CHS* (LNC_000155, LNC_000244 and LNC_002856), *F3′H* (LNC_002411, LNC_003044 and LNC_004152), *FLS* (LNC_002231 and LNC_004260), *F3′5′H* (LNC_001069, LNC_002773 LNC_003359 and LNC_003464) and *DRF* (LNC_002984) (Figure 6). The gene expression levels in qPCR analysis were consistent with the data in transcriptome sequencing. Therefore, the integrated multi-omics analyses highlighted the role of the flavonoid biosynthesis pathway in response to UV-B radiation in *P. nutans*.

## 3. Discussion

The terrestrial ecosystem in Antarctica is experiencing harsh environments, such as enhanced UV radiation, shortage of available water resources and low temperature [28]. In the past two decades, due to the destruction of the ozone layer hole, the surface of Antarctica has been continuously exposed to UV-B radiation [36]. From November to December in 2020, the highest UV irradiance of the Antarctic continent was recorded in the last 20 years at the northern part of the Antarctic Peninsula. Mosses and lichens are the main land plants in Antarctica. In order to cope with the harsh environment, they have evolved a variety of survival strategies from the molecular to cellular level to resist the external environmental pressures. Low dose UV-B radiation could cause gene expression, physiology, metabolite accumulation and morphological changes [37]. However, excessive UV-B radiation can directly damage DNA through the production of cyclobutane pyrimidine dimers (CPDs) [38].

The discovery of non-coding RNAs (ncRNAs) and subsequent elucidation of their functional role have been largely delayed due to the mistaking of the non-protein-coding parts of DNA for “junk DNA”. In recent years, however, in wheat and barley, the identification of lncRNAs has been used to uncover the molecular mechanisms behind developmental and biological stress responses [39]. Comparative analysis of differentially expressed genes and their non-coding RNA partners, long noncoding RNAs and microRNAs, provided valuable insight to gene expression regulation in response to drought stress [40]. In two bread wheat varieties, lncRNA can regulate the ability of the wheat to resist destructive pests [41]. Although a significant number of lncRNAs have been discovered in plants and they may play a variety of roles in controlling different biological processes [10,42], the functions of most lncRNAs are still not fully characterized. Particularly, there are few reports on the roles of lncRNA in response to UV-B radiation. We employed high-throughput sequencing technology to examine the expression profiles of lncRNA and mRNA of the Antarctic moss, *P. nutans*, in order to investigate the function of lncRNA under UV radiation. Through the prediction of coding potential, 5729 lncRNAs were predicted. These lncRNAs can be further classified into lincRNAs, intronic lncRNAs and antisense lncRNAs (Figure 1C). We compared the mRNA and lncRNA sequence features through transcriptome sequencing analysis, such as exon number and transcript length. Compared with mRNA, the predicted lncRNA had a shorter sequence length, fewer exons and lower expression level, which was consistent with previous research results [43,44]. In transcriptome sequencing, the differential expression analysis was widely employed to find genes related to stresses [45,46]. In the present study, 1459 differentially expressed lncRNAs were identified. Of them, 954 DELs were significantly up-regulated and 505 DELs were markedly down-regulated in *P. nutans* under UV-B radiation (Figure 1D). Cis-acting lncRNAs have been demonstrated to activate, repress or otherwise modulate the expression of target genes through various mechanisms [47].

UV-B radiation has deeply affected the terrestrial and aquatic ecosystems on the earth [48]. Low levels of UV-B radiation can mediate plant photomorphogenesis and act as a growth signal, whereas high levels of UV-B radiation will reduce the photosynthetic capacity and produce reactive oxygen species (ROS), thus slowing down plant growth [49,50,51]. UV RESPONSE LOCUS 8 (UVR8) is the UV-B photoreceptor responsible for UV-B photomorphogenesis and acclimation [52]. ELONGATED HYPOCOTYL 5 (HY5) induces the synthesis of RUP (REPRESSOR OF UV-B PHOTOMORPHOGENESIS) proteins, interacts with UVR8, releases constitutively photomorphogenic 1 (COP1), and induces the re-dimerization of UVR8, therefore resulting in environmental adaptation and UV-light-induced photomorphogenesis, including the non-specific pathway of oxidative damage [50,53,54]. The UVR8-COP1-HY5 constitutes the principal part of UV-B radiation signal pathway that has been sufficiently characterized in Arabidopsis. We predicted the target genes of DELs and annotated the target gene to reveal the function of DELs. Through the combined analyses of GO enrichment, we found that DELs and mRNA were both significantly enriched in the plasma membrane, cellular response to oxidative stress, and oxidoreductase activity (Figure 2A, Appendix A). Under UV radiation, plant cell membrane has a vital importance in material transport and osmotic regulation [55]. When plants are exposed to UV radiation, the content of ROS in plants increases significantly. A high concentration of ROS will cause lipid oxidative stress, thus damaging the function of cell membrane [56].

In addition, KEGG pathway enrichment analyses also demonstrated that plant-pathogen interaction and flavonoid biosynthesis were the main enrichment pathways (Figure 2B, Appendix A). In general, UV-B light has a positive effect on disease resistance in many plant–pathogen combinations, mainly through the induction of the production of specialized metabolites [57]. UV-B radiation pretreatment can enhance the resistance to pathogen infection [58]. By screening target genes of DELs, we plotted the plant-pathogen interaction pathways (Figure 3A). In plant-pathogen interaction, disease resistance protein genes (i.e., *PR1*, *RPM1* and *RPS2*) were mostly up-regulated (Figure 3B). Meanwhile, the enhancement of plant-pathogen interaction pathway may be closely related to the accumulation of secondary metabolites such as phenols, flavonoids, anthocyanins, alkaloids and terpenoids in plants under UV-B radiation [59]. UVR8 plays an important role in mediating the effects of UV-B radiation on pathogen resistance, by controlling the expression of the sinapate biosynthetic pathway [53,60]. During the interaction of plant pathogens, plants will produce a large number of polyphenols to resist the invasion of pathogens [35]. In the widely targeted metabolomics analysis, we detected a large number of polyphenol compounds, including flavonoids, phenolic acids and tannins. This further shows that the plant-pathogen interaction plays a very important role in the process of *P. nutans* resistance to UV radiation. At the same time, qPCR analysis was performed to verify the gene expression of *NOS*, *RPM1*, *RPS2*, *EDS*, *KCS* and *WRKY*. Our qPCR analysis results were consistent with the data in transcriptome sequencing (Figure 4). These results showed that UV-B radiation had exerted a great impact on the plant–pathogens interaction pathway.

Flavonoids have attracted considerable attention as antioxidants [61]. From early plants to green plants, throughout evolution, flavonoids which especially glycosylated flavonoids and phenylacylated flavonols protect plants against UV-B damage [62]. Most of flavonoid biosynthesis genes are highly induced by UV-B radiation [63,64], and the content of flavonoids in plants is constantly changing with the prolongation of UV-B radiation exposure [65]. Previous studies have shown that a large number of phenols and flavonoids, such as quercetin, kaempferol and gallic acid, were detected in *Sideroxylon capiri Pittier* exposed to long-term UV-B radiation [66]. Importantly, UV-B radiation enhances the transcripts in the flavonoids biosynthesis pathway, especially the gene expression levels of chalcone synthase (CHS) [59]. Meanwhile, in rice, OsRLCK160 and OsbZIP48 could interact and phosphorylate that, promoting the accumulation of flavonoids to resist external UV-B radiation [67]. Therefore, UV-B radiation could change the biochemical components and produce a large number of secondary metabolites in plants [68]. Transcriptional profiling and physiological analyses reveal the critical roles of the ROS-scavenging system in the Antarctic moss *P. nutans* under UV-B radiation. Previously, we have detected the total content of total flavonoids [69]. We found that the content of flavonoids was markedly increased when plants were exposed to UV-B light. To further investigate the role of flavonoids in the resistance to UV-B stress, we conducted widely targeted metabolomics analysis of *P. nutans*. Metabolome analysis showed that the content of flavonoids was higher in the DCMs (Figure 5A). At the same time, we found that the most significant metabolite was gallic acid (Figure 5B). Gallic acid is a kind of flavonoid, which has a series of biological activities, including antioxidant [70], antibacterial [71] and antifungal [72]. Therefore, flavonoids had a vital importance in the process of *P. nutans* resistance to UV-B radiation. Based on the transcriptome sequencing and metabonomic analysis data, we constructed the co-expression network analysis of DELs and DCMs involved in the flavonoid pathway in response to UV-B radiation (Figure 5C, D). In the flavonoid pathway, most of DELs and DCMs were up-regulated in *P. nutans* under UV-B radiation. We further employed the qPCR technique to verify the gene expression of DELs (*C4H*, *CHS*, *CHI*, *FLS*, *DFR*, *CCoAOMT*, *F3′H* and *F3′5′H*) participating in flavonoids biosynthesis, and the qPCR results were consistent with the data of transcriptome sequencing (Figure 6).

It is worth mentioning that the biosynthesis of flavonoids under UV-B radiation plays a positive effective role in plants fighting against pathogen infection [59]. Previous studies have shown that UV-B-induced defense against *Botrytis cinerea* depends on UVR8, which may be caused by the increase of syringyl type lignin [60]. In addition to lignin, various phenolic compounds have been observed to have antibacterial and antifungal activities [73]. A large number of antibacterial compounds were significantly up-regulated, including quercetin and kaempferol classes of flavonoids. Therefore, flavonoids produced by UV-B radiation may play a positive role in plant-pathogen interaction [74,75]. In conclusion, our results indicated that lncRNAs enhance the UV radiation resistance of Antarctic mosses by targeting the regulation of plant-pathogen interaction and flavonoid synthesis pathways. Most importantly, we found that a large number of DELs and DCMs were involved in the flavonoid synthesis pathway through the integrated multi-omics analyses. These results will lay a foundation for the study of lncRNAs in UV-B radiation.

## 4. Experimental Procedures

### 4.1. Plant Samples and UV-B Radiation Treatments

The plant samples of *Pohlia nutans* strain, LIU, were collected in March 2014 near the Great Wall Station on the Fildes Peninsula in Antarctica (S62°13.260′, W58°57.291′). Plants were cultured on the culture medium that was composed of the Pindstrup substrate (Pindstrup Mosebrug A/S, Ryomgaard, Denmark) and local soil, with the light conditions of 16 °C, 50 µmol photons·m^−2^·s^−1^, and 16 h light/8 h dark light cycle. The UV light, produced by the Philips T8 TLD36W/54-765 fluorescent tubes under these circumstances, was 0.09 mW/cm^2^.

We conducted the UV-B radiation treatment by using two Philips TL20W/01RS narrowband UV-B tubes that were placed above the plants, as previously mentioned [76]. The average UV-B irradiance was 0.30 mW/cm^2^, detected by using a UV-340A UV Light Meter (Lutron Electronic Enterprise, Taibei, the Taiwan region). The white light field was supplemented with two Philips T8 TLD36W/54-765, and photosynthetically active radiation was measured to 13.5 µmol photons·m^−2^·s^−1^ (1000 lux), using an LX-101A Light Meter (Lutron Electronic Enterprise, Taibei, the Taiwan region). The transcriptome sequencing was conducted for *P. nutans* treated with UV-B radiation for 0 h (control group) and 18h (UV-B group), to identify the genes responsible for signal transduction and sensing. We used quantitative real-time RT-PCR to corroborate the transcriptome sequencing results on plants that had been exposed to UV-B radiation for 0 h, 6 h, 18 h, and 36 h (i.e., CK, UV-B6h, UV-B18h and UV-B36h), respectively. For widely targeted metabolomics analysis, *P. nutans* were treated with UV-B light for 0 h (control group), and 3 days (UV-B group), respectively. The experiments were conducted with three biological replicates. The green gametophytes were gathered, quickly frozen in liquid nitrogen, and kept at −80°C for storage.

### 4.2. RNA Extraction and Library Construction

For each sample, we ground 0.2 g of moss gametophyte into powder, and extracted total RNA with the TRIzol reagent (Invitrogen, Carlsbad, CA, USA). We performed a series of assessments on the quality of RNA samples. By using the RNA Nano 6000 Assay Kit and the Agilent Bioanalyzer 2100 system, we determined the RNA integrity (Agilent Technologies, Santa Clara, CA, USA). In addition, the Nanodrop 2000 Spectrophotometer was used to assess the purity of the RNA (Thermo Fisher Scientific, Waltham, MA, USA). The concentration of total RNA was accurately measured using Qubit 2.0 fluorochrome (Thermo Fisher Scientific, Waltham, MA, USA). Then, rRNA was removed from the obtained high-quality total RNA samples using the Ribo-Zero™ rRNA Removal Kit (Illumina, San Diego, CA, USA; removing rRNA can maximize the retention of lncRNA containing the polyA tail). RNA was fragmented into 250–300 bp-long pieces. Using a template of short RNA fragments, six random hexamers were used to synthesized cDNA. Then, two-stranded cDNA was synthesized by adding buffers, dNTP (dUTP, dATP, dGTP and dCTP) and DNA polymerase I. Using AMPure XP beads, the double-stranded cDNA was purified (Beckman Coulter, Brea, CA, USA) and then underwent PCR enrichment to obtain strand-specific cDNA libraries. We established a total of 6 cDNA libraries. Subsequently, these libraries were sequenced using the Illumina NovaSeq 6000 platform (Illumina, San Diego, CA, USA).

### 4.3. Read Mapping to the Reference Genome

We generated the clean data by eliminating adapter sequences, reads containing poly-N and low-quality reads from the raw data, using the FastQC tools [77]. Clean reads were then aligned to the *P. nutans* reference genome [78] using HISAT2 [31]. Using StringTie [79], the lncRNA transcriptome was assembled based on the reads mapped to the *P. nutans* reference genome [78]. The GffCompare tool was used to annotate the assembled transcripts. We set a series of strict conditions to screen the spliced transcripts, and finally obtain the lncRNA. First, we selected the transcript with a length greater than or equal to 200 bp. Then we used CuffCompare to filter transcripts with overlapping regions of mRNA and other non-coding RNA (i.e., rRNA, and tRNA) in the known database. If there was a lncRNA annotation in the database (the lncRNA in the database that overlaps with the exon region of this splicing transcript), we would take the lncRNA as the known lncRNA. For novel lncRNA, we used CPC2 (https://github.com/biocoder/CPC2, accessed on 12 July 2022) [80], CNCI (https://github.com/www-bioinfo-org/CNCI, accessed on 12 July 2022), and PLEK (https://sourceforge.net/projects/plek/, accessed on 12 July 2022) softwares to predict the coding potential of the transcripts, and took the intersection of the predicted results [81,82]. We selected the transcripts without matching as the final lncRNA data set, by comparing with miRBase and Rfam databases. Finally, different kinds of lncRNAs were further identified by CuffCompare (i.e., lincRNA, intron lncRNA and antisense lncRNA) (CuffCompare 2.2.1, http://cole-trapnell-lab.github.io/cufflinks/manual/, accessed on 12 July 2022).

### 4.4. Differential Expression Analysis and Functional Annotation

We employed the StringTie software to calculate FPKM (fragments per kilobase of transcript per million fragments mapped) and determine the expression level of lncRNA and mRNA [79]. We then conducted the differential expression analysis using the DESeq R software [83]. The resulting *p*-value was corrected by multiple hypothesis tests to calculate the FDR value [84]. The FDR was calculated using the corrected difference significance of *p*-value. The differentially expressed lncRNAs (DELs) were identified using the thresholds of an absolute value of log_2_(fold-change) > 1 and an adjusted FDR < 0.05. The differentially expressed genes (DEGs) were identified using the thresholds of an absolute value of log_2_(fold-change) ≥ 1 and an adjusted FDR < 0.05. According to the positional link between lncRNA and target genes, cis-target genes for lncRNAs were primarily anticipated. We identified nearby genes as the cis-target genes of lncRNA within 100 kb upstream and downstream of lncRNA. LncRNA trans-target genes were predicted by lncRNA and mRNA expression correlation analysis or co-expression analysis methods. The correlation between lncRNA and mRNA between samples was analyzed by the Pearson’s correlation coefficient method, and the mRNA with an absolute correlation value > 0.95 and *p*-value < 0.01, was used as trans-target genes of lncRNA.

We performed Gene Ontology (GO) and Kyoto Encyclopedia of Genes and Genomes (KEGG) pathway enrichment analyses on target genes for cis- and trans-regulation of DELs, as well as DEGs. GO enriched terms were found using the Blast2GO software program [85]. DELs and DEGs were considered significantly enriched in regards to GO, with a corrected FDR < 0.05. The statistical enrichment of DELs and DEGs among the KEGG pathways was tested using the KOBAS software [86].

### 4.5. Widely Targeted Metabolomics Analysis

The moss gametophytes of the control group and UV-B treatment group were collected and used for UPLC-MS/MS analysis. A standard procedure of widely targeted metabolomics technique was used to extract the samples, identify the metabolites, and quantify their contents [87,88,89]. The “widely targeted metabolomics technique” was developed based on multiple reaction monitoring (MRM), which is a new integrated technique for the large-scale detection, identification, and quantification of metabolites in a high throughput manner [62,90]. Principal component analysis (PCA), orthogonal projections to latent structure-discriminant analysis (OPLS-DA), and hierarchical clustering analysis (HCA) were performed using the R package (www.r-project.org/, accessed on 12 May 2022) to examine the accumulation of metabolites caused by UV-B radiation. To standardize the data, the various metabolite changes (fold-change) were log-transformed to normalize the data. The relative importance of each metabolite was calculated using the variable importance in projection (VIP) values from the OPLS-DA model. As differentially accumulated metabolites, metabolites with fold-change ≥ 2 or fold-change ≤ 0.5 and VIP ≥ 1 were considered as differentially changed metabolites (DCMs).

### 4.6. Quantitative Real-Time RT-PCR Analysis

To detect the accuracy of gene expression analysis in transcriptome sequencing, we evaluated the results using quantitative real-time RT-PCR techniques (qPCR). Using liquid nitrogen, plant samples were pulverized into a powder. Using the TransZol Up Plus RNA Kit, RNA was extracted (TransGen, Beijing, China). In order to determine the expression levels of genes involved in plant-pathogen interaction and flavonoid synthesis, GAPDH was verified as the reference gene and gene-specific primers were designed for qPCR analysis. The positive primer sequence of GAPDH was: AGGAAGGACTCGCCTCTGGAAG; the reverse primer sequence was: CGATACTGATGCCCGTCGTTGCC. In Appendix A, there was a list of the gene-specific primers. The Perfect Start^®^ Green qPCR Super Mix Kit (TransGen, Beijing, China) was used in the qPCR experiments, which were performed on the Light Cycler^®^ 96 Instrument (Roche, Basel, Switzerland). The cycling time was 2 min at 95 °C, followed by 45 amplification cycles (94 °C 5 s, 60 °C 15 s, 72 °C 10 s). For each template, the reaction was carried out three times. The experiment was repeated three times. The relative gene expression levels were calculated using the Ct(2^−ΔΔCt^) method [91].

### 4.7. Statistical Analysis

Data were presented as the mean ± standard error (SEM) from three independent experiments. Results were statistically compared between different groups for quantitative analysis, and the one-way ANOVA test was used to determine statistical significance (ns *p* > 0.05, * *p* < 0.05, ** *p* < 0.01). All experiments were repeated at least three times and the measurements were done in triplicates.

## 5. Conclusions

We conducted an integrated multi-omics analysis to reveal global properties of the Antarctic moss *P. nutans* under UV-B radiation. We found that the target genes of differentially expressed lncRNAs (DELs) were significantly enriched in plant-pathogen interaction and the flavonoid synthesis pathway, and most of the target genes were up-regulated. Based on the transcriptome sequencing and metabonomic analysis data, we constructed the co-expression network analysis of DELs and differentially changed metabolites (DCMs) involved in the flavonoid pathway and plant-pathogen interaction in response to UV-B radiation. Our results provide insights into the regulatory network of lncRNA under UV-B radiation, and the adaptation of Antarctic moss to the polar environments.

## Figures and Tables

**Figure 1 ijms-24-05757-f001:**
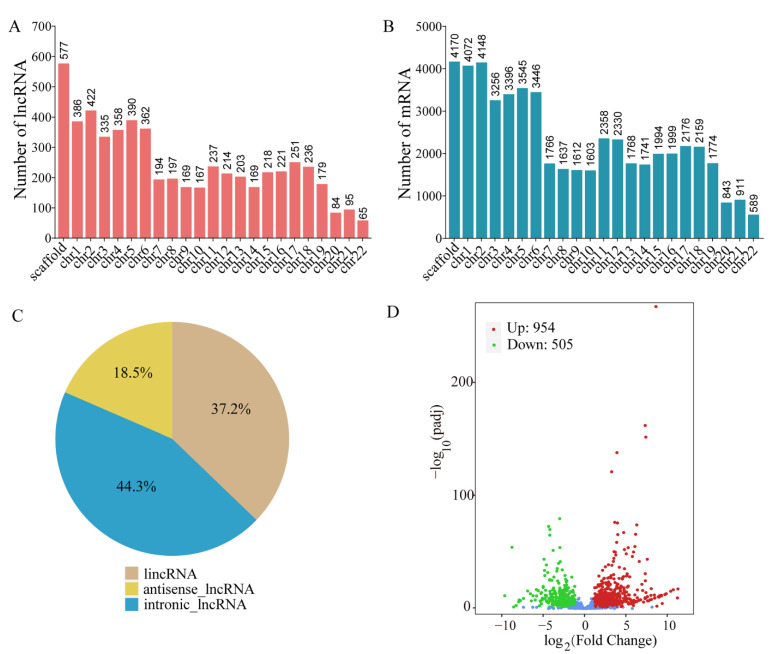
Transcriptome sequencing of the Antarctic moss, *Pohlia nutans*, under UV-B light. (**A**,**B**) Distribution characteristics of lncRNA and mRNA on chromosomes. chr1–22 corresponded to 22 pairs of chromosomes. (**C**) The classification of all lncRNAs of *P. nutans*. (**D**) The volcano plot showing the DELs between UV-B radiation group and the control group. The *X*-axis indicates fold-change of gene expression (threshold, |log_2_ (Treat/Control)| > 1), while the *Y*-axis means the statistically significant level (threshold, q-value < 0.005). Blue dots represent lncRNA detected but not significantly different.

**Figure 2 ijms-24-05757-f002:**
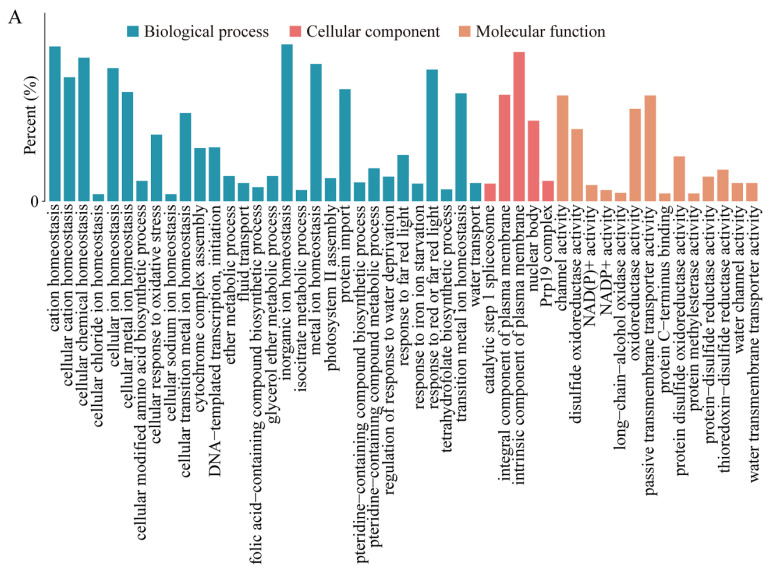
Functional annotation of the cis-target genes of DELs. (**A**) GO enrichment analysis of the cis-target genes of DELs. (**B**) KEGG pathway enrichment of the cis-target gene of DELs. Rich factor represents the ratio of the number of the target gene of DELs to the total number of annotated genes in this pathway.

**Figure 3 ijms-24-05757-f003:**
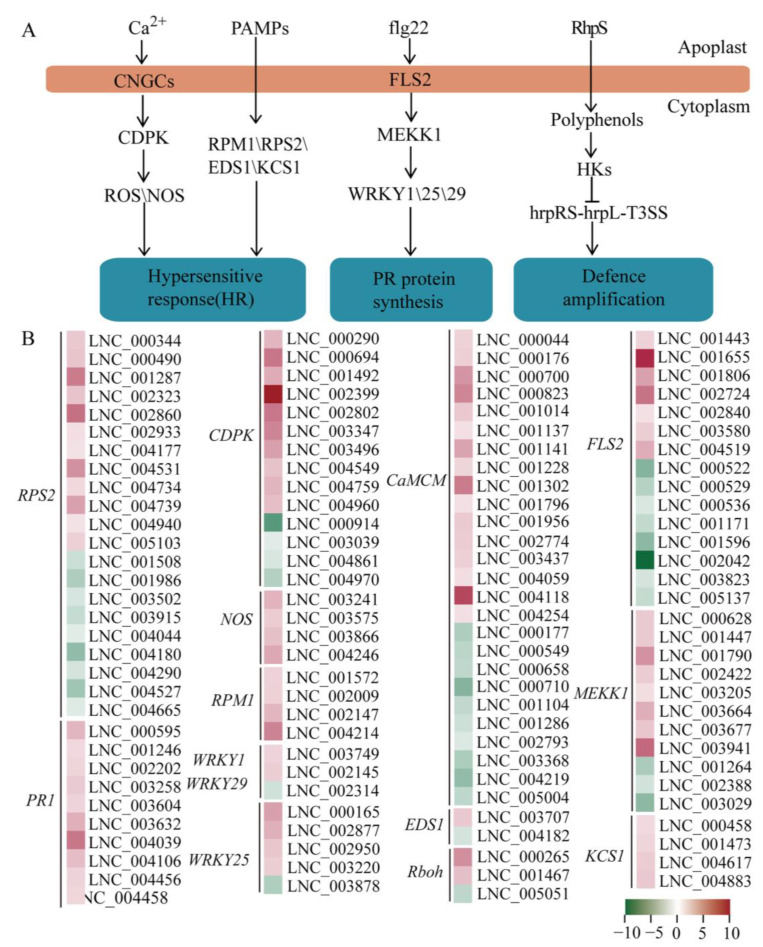
The pathway of plant-pathogen interaction. (**A**) A proposed model summarizing the main pathways for plant-pathogen interaction under UV-B radiation. (**B**) DELs involved in the pathway of plant-pathogen interaction were up-regulated under UV-B radiation.

**Figure 4 ijms-24-05757-f004:**
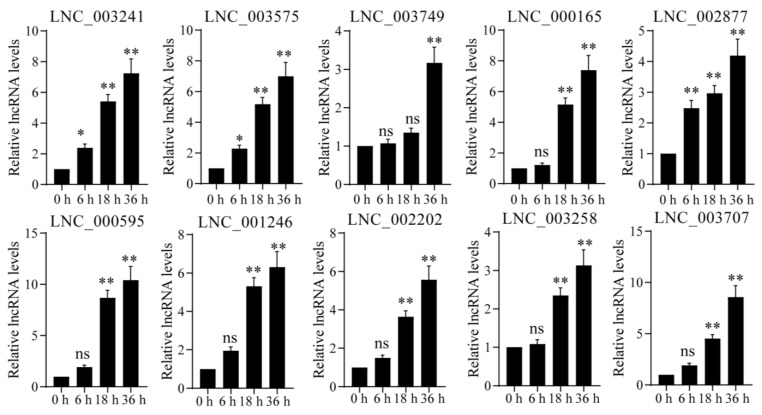
Key DELs of the plant-pathogen interaction pathway were up-regulated under UV-B radiation. The gene expression levels were analyzed by quantitative RT-PCR analysis; the *Y*-axis indicates the relative expression level; the *X*-axis indicates UV-B treatment time (h); The data were calculated from three biological replicates. Vertical bars are the means ± SEM. Significant difference (ns *p* > 0.05, * *p* < 0.05, ** *p* < 0.01). The target genes of these DELs were nitric oxide synthase (LNC_003241 and LNC_003575), WRKY transcription factor 1 (LNC_003749), WRKY transcription factor 25 (LNC_000165 and LNC_002877), pathogenesis-related protein 1 (LNC_000595, LNC_003258, LNC_001246 and LNC_002202), enhanced disease susceptibility 1 protein (LNC_003677), *RPM1*, disease resistance protein RPM1 (LNC_001572, LNC_002009, LNC_002147 and LNC_004214), 3-ketoacyl-CoA synthase (LNC_000458 and LNC_001473), and disease resistance protein RPS2 (LNC_000344, LNC_00490, LNC_001287 and LNC_002323).

**Figure 5 ijms-24-05757-f005:**
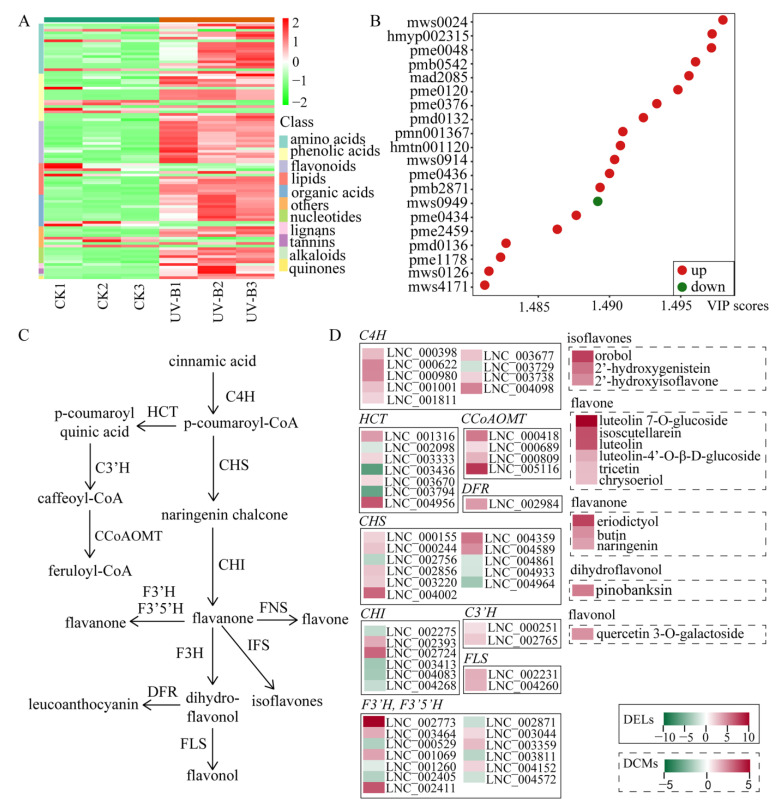
Integrated analyses of transcriptome and metabolome of flavonoid synthesis pathway under UV-B radiation. (**A**) Heatmap of differentially changed metabolites (DCMs) sorted by metabolite classes in *Pohlia nutans* after 3 d of UV-B treatment. (**B**) The VIP scores of the top 20 DCMs between the two groups. (**C**) Integrated transcriptome and metabolome analyses showed that flavonoid biosynthesis might contribute the resistance of *P. nutans* against UV-B radiation. (**D**) The block of each DEL and DCM indicated the log_2_ (foldchange) of this DEL and DCM between the control group and the UV-B group. DELs labeled in solid line box and DCMs showed in dotted box.

**Figure 6 ijms-24-05757-f006:**
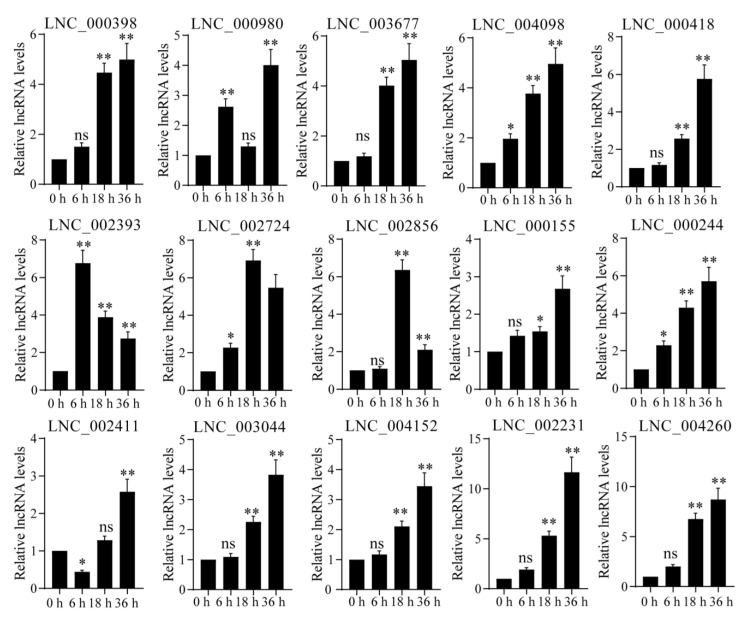
Key DELs of the flavonoid synthesis pathway were up-regulated after UV-B treatment. The gene expression levels were analyzed by quantitative RT-PCR analysis; the *Y*-axis indicates the relative expression level; the *X*-axis indicates UV-B treatment time (h); The data were calculated from three biological replicates. Vertical bars are the means ± SEM. Significant difference (ns *p* > 0.05, * *p* < 0.05, ** *p* < 0.01). The target genes of these DELs were chalcone synthase (LNC_000398, LNC_000980, LNC_003677 and LNC_004098), caffeoyl-CoA O-methyltransferase (LNC_000418), chalcone isomerase (LNC_002393 and LNC_002724), chalcone synthase (LNC_000155, LNC_000244 and LNC_002856), flavonoid 3′-hydroxylase (LNC_002411, LNC_003044 and LNC_004152), flavonol synthase (LNC_002231 and LNC_004260), flavonoid 3′, 5′-hydroxylase (LNC_001069, LNC_002773 LNC_003359 and LNC_003464), and flavanone 4-reductase (LNC_002984).

**Table 1 ijms-24-05757-t001:** Summary of RNA-seq data for the control (CK) and UV-B treatment groups of *Pohlia nutans*.

Sample	Raw Reads	Clean Reads	Clean Base (G)	Error Rate (%)	Q_20_ (%)	Q_30_ (%)	GC Content (%)
CK1	79,930,786	79,469,444	11.92	0.03	96.3	90.14	44.02
CK2	99,222,926	98,500,292	14.78	0.03	96.21	89.96	43.84
CK3	90,289,934	88,589,850	13.29	0.03	96.74	91	44.94
UV-B1	87,048,974	86,334,606	12.95	0.03	96.43	90.38	45
UV-B2	84,624,622	83,986,726	12.6	0.03	96.77	91.04	44.34
UV-B3	96,313,384	95,201,204	14.28	0.03	96.95	91.4	45.38

## Data Availability

The RNA sequencing data generated in our study were submitted to the National Genomics Data Center (NGDC, https://ngdc.cncb.ac.cn, accessed on 24 January 2023) under the accession number of CRA006053.

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
