# Peer review of "Genome-Wide Analysis of Long Non-Coding RNAs Related to UV-B Radiation in the Antarctic Moss Pohlia nutans"

_ijms, 2023, doi:10.3390/ijms24065757_

Round 1

Reviewer 1 Report

I have several major concerns for the manuscript.  

1. The plant pathogen interaction and flavonoid synthesis are the main enrichment pathway. The 

DELs target genes  the differentially change metabolites  related to flavonoid synthesis were detected and analyzed  using  integrated multi-omics methods.

While, for the plant pathogen interaction pathway, only  the results of transcriptomic and gene expression pattern analysis were demonstrated. 

2.A large number of target genes of DELs related to flavonoid synthesis were identified and briefly described , while, the infomration such as the key DELs and their target flavonoid synthesis related genes were missing.

3. It is all know that flavoids play roles in the scavenging of ROS caused by UV or other abiotic stress, was the total flavonoids content  of Antarctic moss Pohila nutans under UV radiation measured?

Reviewer 2 Report

This manuscript identifies and performs functional analysis of lncRNAs from UV-B radiation treatment on Pohlia nutans through transcriptome. Functional enrichment analysis revealed its association with flavonoid biosynthesis and plant-pathogen interactions. The combination with metabolomic analysis also increased the credibility of the analysis. The study of manuscripts is interesting and valuable, but the overall content of the manuscript is relatively simple and could be made more substantial by adding more sections. The manuscript could be enriched by elaborating the regulatory network of lncRNAs and mRNAs, other modes of regulation of lncRNAs and mRNAs (e.g. trans-regulation and base complementary pairing), or by adding further analysis of the association between DELs and DCMs.

Main problems:

1.         The identification and characterization of lncRNAs in the first and second parts of the results can be integrated, and some of the figures that are not very important can be moved to the supplementary figures to highlight the important content in the later part of the manuscript.

2.         In the functional enrichment analysis in the third part of the manuscript, lncRNAs were not very significantly enriched in the flavonoid biosynthesis pathway compared to other pathways with higher q-values for both GO and KEGG enrichment analysis. Considering the enrichment results of mRNA, there are many enrichment pathways for the intersection of the mRNAs and lncRNAs (Fatty acid elongation, Biosynthesis of cofactors, Amino sugar and nucleotide sugar metabolism, phenylpropanoid biosynthesis, etc.), why pick the flavonoid biosynthesis and plant-pathogen interactions for later studies?

3.         The enrichment results of lncRNA and mRNA can be shown by heatmap (-log10(q values) as the value of the heatmap) so that the reader can more easily find the difference of enrichment and the corresponding significance, and also make the manuscript more compact.

4.         In the introduction, the authors mention that the analysis of lncRNA is divided into cis and trans interactions, why the analysis of trans is not performed in the manuscript?

5.         The association analysis between differentially expressed DELs and DCMs was poorly described, and the manuscript mainly mentioned that DELs targeted flavonoid-related genes and flavonoids in DCMs accounted for 20% of the total metabolites. The association analysis was not performed between DELs and DCMs, and the association analysis between DELs and DCMs would make the article more sufficient.

6.         Plant-pathogen interactions are also an important part of the manuscript and in the discussion the authors mention that related metabolites may be associated with them. However, it is not shown in the results section, whether DCMs affecting plant-pathogen interactions are present and how they change?

7.         The manuscript primarily shows changes in lncRNAs, whether DEGs interacting with flavonoid biosynthesis and plant-pathogenic interactions also changed?

8.         The article used CPC2, CNCI, and PLEK for lncRNA screening, whether the lncRNAs obtained were screened against databases such as Rfam and miRbase to remove other types of ncRNAs? For abiotic stress treatment,1459 lncRNAs seem to be much, how many DEGs exist after UV-B radiation treatment?

Minor problems:

1.         The mapping rate should be shown in the data quality.

2.         The DEGs should be given in the supplementary table.

Reviewer 3 Report

This manuscript deals with a multi-omics investigation to find insights into the regulatory function of lncRNA under UVB radiation for an Antarctic moss. Results from RNA sequencing and metabolomis are solid. The work is factual but would serve as the basis of novel research. I expect that this study will contribute to understanding the adaptation mechanism to UVB radiation.

I have only some minor points.

overall manuscript, UV instead of ultraviolet.

line 70, a reference to the sentence describing the distribution of Antarctic plants is required.

line 135, (Figure 2A and Table S2) instead of (Figure 2A).

lines 163-165, the description does not match Figure 3A.

Figure 5A, displaying the level of change in expression for each component will make the figure more informative.

line 353, Pittier, Roman font

line 380, positive instead of positively

line 388, please include the history of Pohlia nutans subsp. LIU. I could not find any information of the subspecies. Have the authors deposited the specimen to the herbarium? Why is it Pohlia nutans subsp. LIU instead of Pohlia nutans or Pohlia nutans strain LIU?

line 483, provide the primer sequence or a reference for GAPDH.

Table S4, a sheet name in English.

Round 2

Reviewer 2 Report

The authors solved most of the problems, but there is still one minor point. In the third point, (-log10(q values) is not the main problem. I think the article Figure 2 to Figure 4 is too similar in presentation. How to change the presentation or put part of figure in the supplement can better highlight the content of the later co-analysis and make the article more compact.

Author Response

Thanks for your comments. We have moved the figures (i.e.,Figure 3 and Figure 4) that are not very important to the supplementary figures.